# Calcium-Dependent Interplay of Lithium and Tricyclic Antidepressants, Amitriptyline and Desipramine, on *N*-methyl-D-aspartate Receptors

**DOI:** 10.3390/ijms232416177

**Published:** 2022-12-19

**Authors:** Sergei I. Boikov, Dmitry A. Sibarov, Yulia D. Stepanenko, Tatiana V. Karelina, Sergei M. Antonov

**Affiliations:** Sechenov Institute of Evolutionary Physiology and Biochemistry of the Russian Academy of Sciences, Torez pr. 44, Saint-Petersburg 194223, Russia

**Keywords:** amitriptyline, desipramine, clomipramine, lithium, NMDA receptors, calcium, tricyclic antidepressants, patch-clamp, ion channels

## Abstract

The facilitated activity of *N*-methyl-D-aspartate receptors (NMDARs) in the central and peripheral nervous systems promotes neuropathic pain. Amitriptyline (ATL) and desipramine (DES) are tricyclic antidepressants (TCAs) whose anti-NMDAR properties contribute to their analgetic effects. At therapeutic concentrations <1 µM, these medicines inhibit NMDARs by enhancing their calcium-dependent desensitization (CDD). Li^+^, which suppresses the sodium–calcium exchanger (NCX) and enhances NMDAR CDD, also exhibits analgesia. Here, the effects of different [Li^+^]s on TCA inhibition of currents through native NMDARs in rat cortical neurons recorded by the patch-clamp technique were investigated. We demonstrated that the therapeutic [Li^+^]s of 0.5–1 mM cause an increase in ATL and DES IC_50_s of ~10 folds and ~4 folds, respectively, for the Ca^2+^-dependent NMDAR inhibition. The Ca^2+^-resistant component of NMDAR inhibition by TCAs, the open-channel block, was not affected by Li^+^. In agreement, clomipramine providing exclusively the NMDAR open-channel block is not sensitive to Li^+^. This Ca^2+^-dependent interplay between Li^+^, ATL, and DES could be determined by their competition for the same molecular target. Thus, submillimolar [Li^+^]s may weaken ATL and DES effects during combined therapy. The data suggest that Li^+^, ATL, and DES can enhance NMDAR CDD through NCX inhibition. This ability implies a drug–drug or ion–drug interaction when these medicines are used together therapeutically.

## 1. Introduction

*N*-methyl-D-aspartate receptors (NMDARs), which belong to a large family of glutamate-activated ionotropic receptors, participate in CNS functioning both in health and neurological and neurodegenerative diseases [1,2]. In particular, elevated NMDAR expression in the CNS and the dorsal root ganglia [3] enhances pre- and postsynaptic NMDAR activity [4] and promotes neuropathic pain. Activation of NMDARs also negatively modulates the capacity of opioids to alleviate inflammatory pain [5]. This is the reason why NMDAR antagonists, broadly known as neuroprotectors, are often used to diminish excessive receptor activation to suppress hyperalgesia [6,7]. In particular, Mg^2+^ as a voltage-dependent blocker of NMDARs inhibits central sensitization and thereby decreases pain hypersensitivity [8]. Tricyclic antidepressants (TCAs), amitriptyline (ATL), and desipramine (DES) are widely used to treat neuropathic pain. ATL and DES anti-NMDAR properties are thought to govern TCA analgetic effects [9], while TCA-induced monoamine uptake inhibition mediates antidepressant action [10]. At concentrations that strongly exceed the therapeutic ones, ATL and DES induce Ca^2+^-independent and voltage-dependent channel block of NMDARs [11,12]. At therapeutic concentrations, however, the NMDAR channel block by these compounds is negligible, but the effects on NMDARs still persists utilizing the mode of NMDAR inhibition that is dependent on extracellular Ca^2+^ [11,12]. In other words, increasing extracellular calcium concentration ([Ca^2+^]) lowers the IC_50_ for NMDAR inhibition by ATL [11] and DES [12]. This type of inhibition involves TCA-induced enhancement of NMDAR Ca^2+^-dependent desensitization (CDD) [11,12,13], which requires Ca^2+^ entry via activated channels and Ca^2+^ accumulation in close proximity to NMDAR C-terminal intracellular domains [14,15], for review [16]. As a consequence of this, pharmacological agents affecting Ca^2+^ transmembrane transporting mechanisms may interfere with the NMDAR CDD process.

One such an agent is lithium, which causes a substrate inhibition of many sodium-dependent neurotransmitter transporters, including those involved in glutamatergic transmission. In particular, the substitution of Li^+^ for Na^+^ in the external solution allowed for the reveal of non-quantal glutamate release from synaptic terminals and glia [17,18]. Lithium also inhibits the transport currents of the Na^+^/Ca^2+^-exchanger (NCX) [19,20] that may affect NMDA-elicited currents [21,22]. Instead of NMDAR channel block, the suppression of Ca^2+^ extrusion by NCX inhibitors such as lithium or KB-R7943 results in the enhancement of NMDAR CDD, which represents an indirect action on NMDARs [21]. By blocking the forward mode of the NCX [19], the extracellular Li^+^ increases intracellular [Ca^2+^] and, presumably, may interfere with Ca^2+^-dependent inhibition of NMDARs by TCAs. At high doses, lithium carbonate causes body intoxication [23], but at low doses it is commonly utilized to treat depression and bipolar disorders [24]. During the therapy, Li^+^ may reach 0.5–1 mM in blood plasma [24,25]. Lithium salts are also used in combination with monoamine-acting antidepressants [26,27] for treatment-resistant depression patients.

Taking into account the fact that both Li^+^ and TCAs target a common Ca^2+^-dependent process, which is probably involved in the TCA action on NMDARs, we can assume that a drug–drug or ion–drug interaction between antidepressants and Li^+^ may occur. Considering the lack of information on how therapeutic Li^+^ concentrations ([Li^+^]) may interfere with the TCA action on NMDARs, here the effects of different [Li^+^]s on the TCA inhibition of currents through native di- and triheteromeric NMDARs composed of GluN1, GluN2A, and GluN2B subunits [28,29,30] in rat cortical neurons were investigated. We demonstrate that in the presence of 1–2 mM [Ca^2+^] the therapeutically relevant [Li^+^]s strongly elevate ATL and DES IC_50_s for NMDAR inhibition. Another TCA, clomipramine (CLO), causes inhibition of NMDAR currents, which is not affected by Li^+^. This property coincides with the observation that the open-channel block of NMDARs by CLO is not Ca^2+^-dependent [12].

## 2. Results

### 2.1. Lithium Prevents the Ca^2+^ Dependence of NMDAR Inhibition by ATL

To clarify if Li^+^ can interfere with the Ca^2+^-dependent inhibition of NMDAR by ATL, the concentration–inhibition relationships for ATL effects on NMDARs as a function of extracellular [Li^+^] were studied. With this goal, ATL concentrations ([ATL]) from 1 to 600 µM were sequentially applied with an increment to the steady state of NMDA-activated currents in the presence of 1 or 2 mM Ca^2+^ in Li^+^- or Na^+^-containing bathing solutions (Figure 1A). The increase in [ATL] induced a progressing decrease in the amplitudes of currents. The amplitude values were plotted as a function of [ATL] and the IC_50_ values for ATL inhibition of NMDA-activated currents were measured by fitting the data with the Hill equation. The ATL IC_50_ obtained under control conditions (Na^+^-containing solution) coincides well with the previous data [11] and demonstrates a decrease in ATL IC_50_ with the [Ca^2+^] elevation from 1 to 2 mM (Figure 1B,C; Table 1). The substitution of Li^+^ for Na^+^ in the bathing solution increased ATL IC_50_ for both [Ca^2+^]s under study (Table 1). In contrast to the Na^+^-containing solution, in the Li^+^- containing solution, the dependence of ATL IC_50_ on [Ca^2+^] disappeared since there was no significant difference between ATL IC_50_ values in 1 and 2 mM [Ca^2+^]s (Figure 1B,C; Table 1). This contradiction forced us to perform further investigation. In particular, the effects of partial substitution of Li^+^ for extracellular Na^+^ on the ATL inhibition of NMDARs were tested.

### 2.2. Quantitative Estimation of Li^+^ Effect on ATL and DES Inhibition of NMDAR Currents

In animal models, Li^+^ exhibits analgetic properties against neuropathic pain within the blood plasma concentration range of 0.6–1.2 mM [31]. This raises the question of whether therapeutic [Li^+^]s are sufficient to affect the ATL or DES inhibition of NMDARs. In the following experiments performed at 2 mM [Ca^2+^], we investigated the effects of the partial substitution of Li^+^ for extracellular Na^+^ on ATL inhibition of NMDAR currents. The data obtained in the bathing solutions containing 1, 7, 18, 30, and 72 mM [Li^+^]s were pulled to plot concentration–inhibition relationships for ATL effect on NMDARs (Figure 2A,B). An increase in [Li^+^] caused a progressive increase in IC_50_ for the ATL inhibition of NMDAR currents. The lithium effect on the IC_50_ reached the maximal value at 18 mM Li^+^ and successive [Li^+^] increase did not further change ATL IC_50_ (Figure 2D; Table 2). The dependence of ATL IC_50_ on [Li^+^] revealed the half-maximal effective [Li^+^] concentration (Li^+^ EC_50_) of 1.4 ± 0.57 mM, which is similar to therapeutic [Li^+^]s [25,31], so that even 1 mM Li^+^ causes a ~10-fold increase in ATL IC_50_ (Table 2).

In addition, the ATL IC_50_ in 18 mM Li^+^ did not differ significantly from those obtained in neurons loaded with BAPTA to prevent the induction of NMDAR CDD (Figure 2D; Table 2). This observation suggests that at 18 mM Li^+^ the Ca^2+^-dependence of ATL inhibition of NMDA-activated currents has disappeared. In the presence of [Li^+^] ≥18 mM, therefore, the experimental value of ATL IC_50_ reaches ~100 µM, which corresponds well to those found previously for the channel block of NMDAR by ATL [11]. From these measurements, we can conclude that at therapeutic concentrations both Li^+^ and ATL enhance the NMDAR CDD and may compete for similar Ca^2+^-dependent molecular targets. This hypothesis became evident from the experiments in which 10 µM ATL effects on NMDAR currents were examined before and after the addition of 0.5 mM Li^+^. In 2 mM Ca^2+^-containing solution both 0.5 mM Li^+^ and 10 µM ATL caused similar inhibition of NMDAR steady-state currents (Figure 2E). Moreover 10 µM ATL did not inhibit the NMDAR current in the presence of Li^+^. An observation that their effects are not additive suggests an ion–drug competition for the same molecular target, which presumably is responsible for the induction of NMDAR CDD.

DES is another TCA, which induces Ca^2+^-dependent inhibition of NMDARs [12] at therapeutic concentrations of ~1 µM. Similar to ATL, experiments testing the effects of partial substitution of Li^+^ for extracellular Na^+^ on DES inhibition of NMDA-activated currents were performed. Because in the case of DES the Ca^2+^ dependence of NMDAR inhibition develops at lower [Ca^2+^]s compared to ATL [11,12], we used [Ca^2+^] = 1 mM for experiments with DES. The data obtained in the presence of 0, 1, and 7 mM [Li^+^]s were plotted to estimate the concentration–inhibition relationships for DES on NMDARs (Figure 3A,B). An elevation of [Li^+^] caused a progressive increase in IC_50_ for DES inhibition of NMDAR currents. The dependence of DES IC_50_ on [Li^+^] revealed the Li^+^ EC_50_ value of 1.3 ± 0.4 mM, which is quite similar to the value found for ATL.

In the presence of 7 mM Li^+^, the IC_50_ for DES inhibition of NMDARs reached maximal value, which differed from DES IC_50_ measured in Na^+^-containing solution, but was similar to those obtained in Na^+^-containing solution in BAPTA-loaded neurons (Figure 3C; Table 1). Therefore, the lack of NMDAR CDD in neurons loaded with BAPTA and the addition in the presence of 7 mM Li^+^ both abolish high-affinity Ca^2+^-dependent inhibition of NMDARs by DES.

It appears that therapeutic concentrations of lithium prevent the calcium-dependent effect of ATL and DES on NMDARs. This raises the question of whether Li^+^ can influence an inhibition of NMDARs by clomipramine (CLO), whose effect on NMDARs is not Ca^2+^ dependent [12].

### 2.3. Lithium Do Not Change Inhibition of NMDARs by CLO

In our previous experiments, obtained in the absence of Li^+^ [11,12], ATL and DES exhibited a profound Ca^2+^-dependence of NMDAR inhibition, suggesting a contribution of CDD in their effects, whereas CLO demonstrated a lack of Ca^2+^ dependence. If Li^+^ may prevent the Ca^2+^-dependent mode of ATL and DES inhibition of NMDARs, then we may predict the lack of Li^+^ effect on the CLO inhibition of NMDARs. To verify this assumption, we studied the effects of the partial substitution of Li^+^ for extracellular Na^+^ on CLO inhibition of NMDAR currents in the presence of 2 mM Ca^2+^ in the bathing solution containing 0 or 7 mM Li^+^. This [Li^+^] induced maximal effects on both DES and ATL NMDAR inhibition (Figure 2C and Figure 3C). The data were pulled to plot the concentration–inhibition relationships for CLO on NMDARs (Figure 4A,B). We observed no significant difference between IC_50_ values of CLO in the absence of Li^+^ and the presence of 7 mM Li^+^ (Figure 4C; Table 2). Obviously, Li^+^ is not able to affect the CLO inhibition of NMDARs. In particular, CLO effects on NMDARs do not exhibit the Ca^2+^-dependence, which is subjected to Li^+^ action.

The observations presented here allow us to suggest that Li^+^ interferes with NMDAR CDD, and may cause an apparent weakening of the Ca^2+^ dependence of ATL and DES effects on NMDARs. In agreement, CLO, as a pure open-channel blocker of NMDARs, is not sensitive to the presence of Li^+^.

## 3. Discussion

In general, the main observations of our investigation could be defined as (i) the Li^+^ effects on NMDAR ATL and DES inhibition are Ca^2+^-dependent, (ii) EC_50_s of Li^+^ effects to increase the IC_50_ of ATL and DES were within 0.8–2.0 mM, which overlaps the therapeutic range of Li^+^ concentrations (0.5–1 mM) found in the blood plasma, and (iii) Li^+^ does not change the parameters of CLO effect on NMDARs, which performs exclusively the open-channel block and is Ca^2+^-independent in contrast to ATL and DES. Overall, this type of pharmacology is consistent with the competitive action of Li^+^ and ATL or DES. 

The observed convergence of the effects of Li^+^ and ATL or DES could be explained by the competition for the same molecular targets, which results in a considerable weakening of the ATL and DES effects on NMDARs. Because NMDAR CDD is the only Ca^2+^-dependent process in the kinetics of the NMDAR activation and the NMDAR inhibition by Li^+^ [21,22], ATL [11], and DES [12] is dependent on extracellular [Ca^2+^], the CDD seems to represent a most likely process that could be targeted by these medicines. Therefore, any molecule involved in the CDD regulation may in principle represent a molecular target for Li^+^ and TCA action. As we previously demonstrated [21,22], for review [16] NMDAR CDD is strongly modulated by the Na^+^/Ca^2+^-exchanger (NCX). The inhibition of NCX by Li^+^ [19,20,32] or by selective inhibitors, such as KB-R7943 [33], prevents the removal of Ca^2+^ entering via activated NMDAR channels. This enhances Ca^2+^ accumulation in the close proximity of the NMDAR intracellular domain and facilitates NMDAR CDD [22]. Hence, in the presence of extracellular Ca^2+^, NCX inhibitors cause a decrease in NMDA-elicited currents, presumably by an indirect enhancement of NMDAR CDD [21,22]. NCX inhibitors do not affect NMDAR currents if NMDAR CDD is prevented by an absence of extracellular Ca^2+^ or in the case of binding of intracellular Ca^2+^ with BAPTA [21,22]. ATL and DES demonstrate similar NMDAR Ca^2+^-dependent inhibition [11,12], which can be related to their effects on NCX [34]. From these experiments, we therefore may suggest that NCX represents a molecular target for Li^+^, ATL, and DES inhibition. 

Experimentally obtained values of Li^+^ EC_50_ that affect ATL and DES IC_50_s overlap with those found in the blood plasma of patients during therapy [25,31]. TCA concentrations observed in the plasma of patients usually do not exceed 1 µM [35], which is close to IC_50_ for ATL and DES Ca^2+^-dependent inhibition of NMDARs at physiological [Ca^2+^]s [11,12]. In our experiments, 1 mM Li^+^ caused an increase in IC_50_ values of ~10 folds and ~4 folds for ATL and DES NMDAR inhibition, correspondently. In the presence of 1 mM Li^+^, therefore, the therapeutic ATL and DES concentrations will become insufficient to considerably inhibit NMDARs that may suggest the existence of some interactions of therapeutic effect in the case of combined therapy.

Our data favor the conclusion that [Li^+^]s above 0.5 mM cause the development of NMDAR CDD so that ATL and DES could not add anything to further induction of CDD. This explains an inability of 10 µM ATL to inhibit NMDA-activated currents in the presence of 0.5 mM Li^+^. In contrast, CLO does not exhibit Ca^2+^ dependence of NMDAR inhibition [12] and its IC_50_ is not altered by Li^+^, suggesting that CLO therapeutic action against neuropathic pain [36,37] is not related to NMDAR inhibition and probably utilizes well-known monoamine uptake inhibition. The phenomenology of Li^+^ action on NMDARs, in general, is similar to those of ethanol and cholesterol [38]. In particular, the high-affinity Ca^2+^-dependent component of ethanol inhibition of NMDARs occurs by CDD enhancement and vanishes when CDD is already fully exacerbated by cholesterol extraction [38].

Taken together, our observations suggest that the competition between Li^+^ ions with ATL and DES occurs at their therapeutically relevant concentrations. TCAs including ATL, DES, CLO, and imipramine are commonly utilized to treat neuropathic pain and depression and can be combined with Li^+^ [26,27] for treatment-resistant depression patients. The proposed analgesic mechanism for TCAs is different from their antidepressant action [39,40,41] but may involve TCA anti-NMDAR effects [42]. This is supported by the observation that neuropathic pain is enhanced by excessive activation of glutamate receptors in the spinal dorsal horn [43,44,45] by Ca^2+^-dependent phosphorylation of NMDARs [46]. Furthermore, the capacity of opioids to alleviate inflammatory pain is negatively regulated by NMDARs [5]. In contrast NMDAR inhibition by Mg^2+^, a physiological voltage-dependent blocker of NMDARs prevents central sensitization and decreases pain hypersensitivity by reducing Ca^2+^ influx [8]. Therefore, ATL and DES anti-NMDAR action at therapeutic concentrations [11,12] may contribute to TCA antinociceptive effects.

Similar antinociceptive effects in neuropathic pain were demonstrated for NCX inhibitors [31,47,48,49]. For example, extracellular Li^+^, which inhibits NCX-mediated Ca^2+^ extrusion, can decrease neuropathic pain at a blood plasma concentration of 0.6–1.2 mM [31]. Both single injection [47] and chronic treatment [48] with lithium chloride alleviated mechanical allodynia. Other NCX inhibitors such as KB-R7943 have a rapid antinociceptive effect against neuropathic pain induced by intracellular Ca^2+^ accumulation in the dorsal root ganglion neurons [49].

Li^+^ inhibition is not specific to NCX and spreads to other Na^+^-dependent transporters, which extends the range of its pharmacological targets in the CNS and complicates the interpretation of Li^+^ effects. TCAs also differ by analgetic properties due to the variability of preferential molecular targets, which may coincide with those of Li^+^. 

The Li^+^-resistant TCA CLO utilizes a different mechanism of action, which suggests the possibility of additive action with Li^+^ during the therapy. A weak CLO effect on NMDARs at therapeutic concentrations of 0.5–4 µM [36,37] may provide a clue as to why CLO is rarely utilized for the treatment of neuropathic pain in contrast to ATL and DES. Synergetic action against neuropathic pain also occurs when the Li^+^-sensitive TCA, ATL, is combined with paracetamol [50], because these two medicines obviously have different molecular targets for their therapeutic action. The similarity of effects of Li^+^, KB-R7943, and Li^+^-sensitive TCAs (ATL, DES) promoting the NMDAR CDD favors their common mechanism of action against neuropathic pain. However, the competitive interaction of these substances on NCXs affecting the NMDAR inhibition implies possible drug–drug or ion–drug interaction, when these medicines are used together for therapy.

## 4. Materials and Methods

### 4.1. Primary Culture of Cortical Neurons

All procedures using animals were performed according to the guidelines of the Federation for Laboratory Animal Science Associations (FELASA) and were approved by the Animal Care and Use Committees of Sechenov Institute. The 17-day pregnant rats were provided by Sechenov Institute Animal Facility. Animals were sacrificed by 1 min CO_2_ inhalation in a plastic container connected to a CO_2_ tank. Fetal brains were used to obtain primary cultures of rat cortical neurons using conventional procedures as described earlier [51]. Neurons were incubated in a “Neurobasal” culture medium supplemented with B-27 (Thermo Fisher Scientific, Waltham, MA, USA) on 7 mm glass coverslips coated with poly-D-lysine. Experiments were performed after 10–14 days in culture [51,52].

### 4.2. Patch Clamp Recordings

Whole-cell currents from cultured rat cortical neurons were recorded using a MultiClamp 700B patch-clamp amplifier, low-pass filtered at 400 Hz, and digitized at the acquisition rate of 20 kilosamples per second using Digidata 1440A and pClamp v10.6 software (Molecular Devices). Solution exchange was performed by means of a fast perfusion system as described earlier [21]. The external bathing solution contained (in mM): 144 NaCl; 2.8 KCl; 1 or 2 CaCl_2_; 10 HEPES, at pH 7.2–7.4, osmolarity 310 mOsm. The pipette solution contained (in mM): 120 CsF, 10 CsCl, 10 EGTA, and 10 HEPES, osmolarity 300 mOsm, with pH adjusted to 7.4 with CsOH. Patch pipettes of 4–6 MΩ were pulled from Sutter BF150-89-10 borosilicate glass capillaries. Experiments were performed at room temperature (22–25 °C). Currents were recorded on neurons voltage clamped at −70 mV. Data are reported without corrections for liquid junction potential, which was −11 mV in our experiments. NMDAR currents were elicited by 100 µM NMDA co-applied with 30 µM glycine as a co-agonist. 

### 4.3. Drugs

Compounds were acquired from Sigma-Aldrich, St. Louis, MO, USA. Particularly NMDA M3262, Glycine G7126, Amitriptyline A8404, Desipramine D3900, Clomipramine C7291.

### 4.4. Analysis of Membrane Currents

To determine the blocking potency of TCAs, NMDA-elicited currents were measured in the absence and presence of different TCA concentrations. Amplitudes of currents measured in the presence of blocker (I_b_) were normalized to the amplitude of maximal currents obtained in control (I_c_). The IC_50_ as the concentration of TCA that causes 50% inhibition of currents and the Hill coefficient (*h*) were estimated by fitting concentration–inhibition curves with the Hill equation: I_b_/I_c_ = 1/(1 + [TCA]*^h^*/IC_50_*^h^*).

### 4.5. Statistical Analysis

Data are shown as representative measurements and mean values ± standard error of the mean (SEM). (*n*) refers to the number of recorded cells. Data pairs were compared using an unpaired Student’s two-tailed *t*-test. Multiple groups were compared using one-way AVOVA and post *hoc* Tukey’s test. Statistical significance is reported in the figures according to the following symbols *, **, ***, and ****, which indicate *p* values below (<) 0.05, 0.01, 0.001, and 0.0001, respectively. In addition, *p* values for each comparison are indicated. Curve fitting was performed using OriginPro software (OriginLab Corp.). IC_50_ values obtained from individual experiments performed under the same experimental conditions were averaged to get mean ± SEM values.

## Figures and Tables

**Figure 1 ijms-23-16177-f001:**
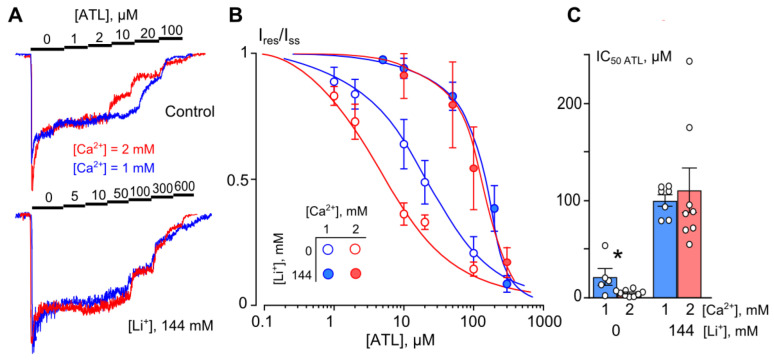
Modulation of amitriptyline (ATL) inhibition of NMDAR currents by Ca^2+^ is prevented by Li^+^. (**A**) Examples of currents activated by 100 μM NMDA + 30 μM Gly recorded at −70 mV in the presence of 1 or 2 mM Ca^2+^ in the bathing solution in the absence of ATL (0) and the presence of rising ATL concentrations ([ATL]) indicated by numbers at the corresponding level of currents (in μM). Currents normalized to the steady-state amplitude without ATL in control and in 144 mM [Li^+^] (full substitution of Na^+^ for Li^+^) are shown. (**B**) Concentration–inhibition curves for ATL of currents activated by 100 μM NMDA + 30 μM Gly recorded at −70 mV in the presence of different [Ca^2+^]s in the bathing solution for control conditions and for 144 mM extracellular [Li^+^]. Symbols show mean values ± S.E.M of the relative amplitudes of currents (I_b_/I_c_) in the presence (I_b_) and absence (I_c_) of different [ATL]s. Solid lines are the fits to the data with the Hill equation. (**C**) The IC_50_s for ATL inhibition of NMDAR currents in control bathing solution and Li^+^-solution, obtained at 1 and 2 mM of extracellular [Ca^2+^]. In control conditions, data for 1 mM [Ca^2+^] differ significantly from those for 2 mM [Ca^2+^] (*, *p* = 0.02, one-way ANOVA). Data from each experiment (symbols) and mean values ± S.E.M. are shown.

**Figure 2 ijms-23-16177-f002:**
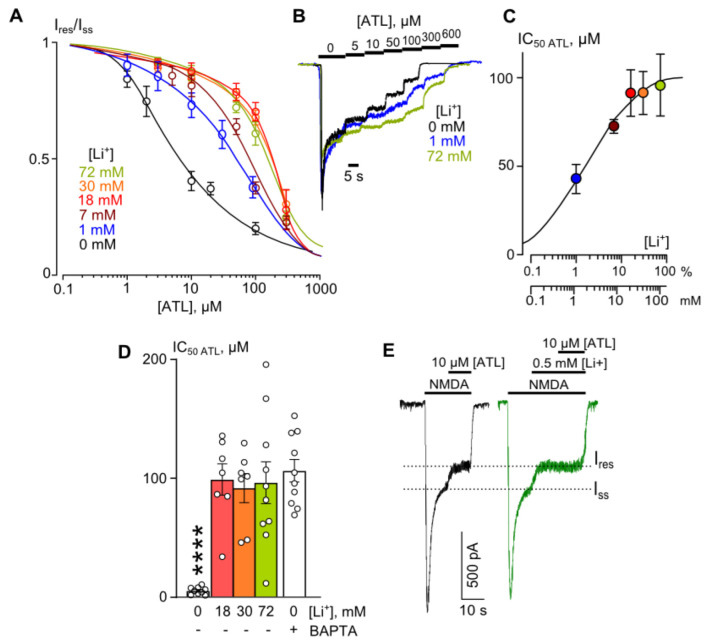
The effect of extracellular lithium on amitriptyline (ATL) inhibition of NMDAR currents in the bathing solution containing 2 mM [Ca^2+^]. (**A**) Dependence of IC_50_ for ATL inhibition of NMDAR currents on extracellular [Li^+^]. Symbols show mean values ± S.E.M of the relative amplitudes of currents (I_b_/I_c_) in the presence (I_b_) and absence (I_c_) of different ATL concentrations ([ATL]). Solid lines are the fits to the data with the Hill equation. (**B**) Examples of currents activated by 100 μM NMDA + 30 μM Gly recorded at −70 mV in the presence of increasing [ATL] in the control bathing solution and in 1 or 72 mM [Li^+^]. [ATL]s are indicated by numbers at the corresponding level of currents (in μM). Currents are normalized to the steady-state amplitude without ATL. (**C**) Dependence of IC_50_ for inhibition of NMDAR currents by ATL on extracellular [Li^+^], obtained from experiments illustrated in panel **A**. The [Li^+^] corresponding to 50% effect on ATL IC_50_ was estimated as 1.4 ± 0.57 mM using Hill equation. (**D**) Mean IC_50_s for ATL inhibition of NMDAR currents in control, in the presence of 18–72 mM [Li^+^], and in neurons loaded with BAPTA. Data in control differ significantly from those in the presence of Li^+^ or in BAPTA-loaded neurons (****, *p* = 0.00001, one-way ANOVA). Data from each experiment (symbols) and mean values ± S.E.M. are shown. (**E**) Currents activated by 100 μM NMDA + 30 μM Gly recorded at −70 mV in a single neuron in the bathing solution containing 2 mM [Ca^2+^] in control and in the presence of 0.5 mM [Li^+^]. 10 μM ATL was applied at the steady state of NMDA-activated currents.

**Figure 3 ijms-23-16177-f003:**
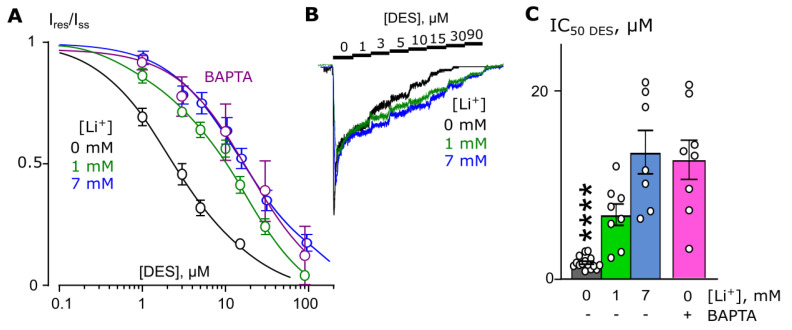
Extracellular lithium effect on desipramine (DES) inhibition of NMDAR currents in the bathing solution containing 1 mM [Ca^2+^]. (**A**) Dependence of IC_50_ for DES inhibition of NMDAR currents on extracellular [Li^+^]. Symbols show mean values ± S.E.M of the relative amplitudes of currents (I_b_/I_c_) in the presence (I_b_) and absence (I_c_) of different DES concentrations ([DES]). Solid lines are the fits to the data with the Hill equation. (**B**) Examples of currents activated by 100 μM NMDA + 30 μM Gly recorded at −70 mV in the presence of increasing [DES] in control bathing solution and in 1 mM or 7 mM [Li^+^]. [DES]s are indicated by numbers at the corresponding level of currents (in μM). Currents are normalized to the steady-state amplitude without DES. (**C**) Mean IC_50_s for DES inhibition of NMDAR currents in control, in the presence of different [Li^+^]s, and in neurons loaded with BAPTA. Data in control differ significantly from those in the presence of 7 mM [Li^+^] or in BAPTA-loaded neurons (****, *p* = 0.00001, one-way ANOVA). Data from each experiment (symbols) and mean values ± S.E.M. are shown.

**Figure 4 ijms-23-16177-f004:**
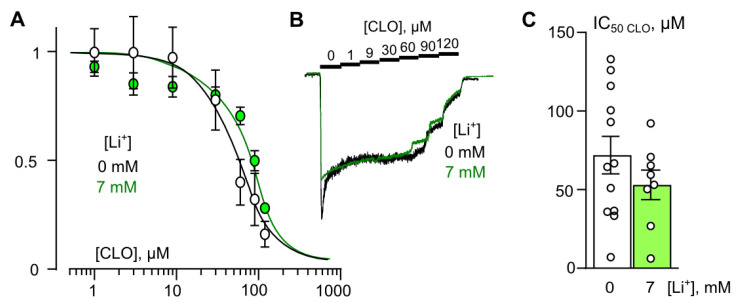
Extracellular lithium effect on clomipramine (CLO) inhibition of NMDAR currents in the bathing solution containing 2 mM [Ca^2+^]. (**A**) Dependence of IC_50_ for CLO inhibition of NMDAR currents on extracellular [Li^+^]. Symbols show mean values ± S.E.M of the relative amplitudes of currents (I_b_/I_c_) in the presence (I_b_) and absence (I_c_) of different CLO concentrations ([CLO]). Solid lines are the fits to the data with the Hill equation. (**B**) Examples of currents activated by 100 μM NMDA + 30 μM Gly recorded at −70 mV in the presence of increasing [CLO] in control bathing solution and in the presence of 7 mM [Li^+^]. [CLO]s are indicated by numbers at the corresponding level of currents (in μM). Currents are normalized to the steady-state amplitude without CLO. (**C**) Mean IC_50_s for CLO inhibition of NMDAR currents in control and in the presence of 7 mM [Li^+^]. No significant IC_50_ difference is observed. Data from each experiment (symbols) and mean values ± S.E.M. are shown.

**Table 1 ijms-23-16177-t001:** An effect of substitution of extracellular Li^+^ for Na^+^ on IC_50_ values for inhibition of NMDAR currents by ATL and DES.

	[Ca^2+^], (mM)	IC_50_ ATL, (μM)	IC_50_ DES, (μM)
Na^+^-solution	1	21.5 ± 8.7 (*n* = 5) * *p* = 0.02	1.75 ± 0.17 (*n* = 15)
	2	5.0 ± 1.0 (*n* = 9)	
Li^+^-solution	1	110 ± 23 (*n* = 8) *** *p* = 0.0001	38.5 ± 3.0 (*n* = 11) ** *p* = 0.0012
	2	100 ± 6 (*n* = 7) *** *p* = 0.0001	

*—the data are significantly different from those obtained at 2 mM [Ca^2+^] in Na^+^-containing solution. **, ***—the data are significantly different from those obtained in Na^+^-containing solution for the corresponding [Ca^2+^]s. Data were compared using one-way ANOVA with post *hoc* Tukey’s test.

**Table 2 ijms-23-16177-t002:** An effect of different [Li^+^]s on the IC_50_ values for inhibition by tricyclic antidepressants of NMDAR currents.

[Li^+^], (mM)	IC_50_ ATL, (μM)	IC_50_ DES, (μM)	IC_50_ CLO, (μM)
0	5.0 ± 1.0 (*n* = 9) **** *p* = 0.00001	1.75 ± 0.17 (*n* = 15) **** *p* = 0.00001	72 ± 12 (*n* = 12)
1	49.0 ± 9.3 (*n* = 11)	6.8 ± 1.1 (*n* = 8)	
7	69.0 ± 3.9 (*n* = 6)	13.4 ± 2.3 (*n* = 7)	53.0 ± 9.3 (*n* = 8)
18	99 ± 13 (*n* = 7)		
30	91 ± 12 (*n* = 7)		
72	96 ± 17 (*n* = 10)		
0 + BAPTA	106.0 ± 9.4 (*n* = 10)	15.0 ± 2.5 (*n* = 8)	

****—the data are different from those obtained in other conditions for specific TCA. Data were compared using one-way ANOVA and post *hoc* Tukey’s test.

## Data Availability

The original contributions presented in the study are included in the article; further inquiries can be directed to the corresponding author.

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
