# Peer review of "Calcium-Dependent Interplay of Lithium and Tricyclic Antidepressants, Amitriptyline and Desipramine, on N-methyl-D-aspartate Receptors"

_ijms, 2022, doi:10.3390/ijms232416177_

Round 1

Reviewer 1 Report

The manuscript addresses a very interesting topic, providing information likely useful for other neuroscientists. Introduction is adequate, results are clearly presented and discussion well structured. Methodological aspects are adequately described, the whole manuscript is well written.

I think a few important points might be clarified and some minor corrections are needed.

Main comments:

It is stated that the therapeutic concentrations of TCAs generally do not exceed 1 µM. Does this apply also to CLO ? Data here presented however show  effects on NMDA currents resulting from concentrations of drugs that are often higher than 1 µM  even in the absence of Li.  I think this point would benefit from more clear explanation especially as regards CLO. It seems not easy for a reader to link the mechanism of action of CLO on NMDA receptor here described (having IC50 about 72 µM, see Table 2) to its therapeutic properties.

MINOR :

Page 3, line 111. I think  “DES” is a mistake,  ATL should be the correct one.

Page 5, line 166. “Fig 2D” is wrong and should be corrected into “Fig 2E”.

Typos: page 5, line 172 : data “were” compared…..

 page 6, line 202  : “7 mm [Li+]”  instead of “72 mM”. 

Author Response

We thank the reviewer for valuable comments. According to the recommendations we have made some changes to the manuscript.

General

In discussion we added information concerning CLO concentrations utilized for therapy, and the mechanisms of CLO action. The following sentences were added:

"In contrast, CLO does not exhibit Ca2+ dependence of NMDAR inhibition [12] and its IC50 is not altered by Li+, suggesting that CLO therapeutic action against neuropathic pain [36,37] is not related to NMDAR inhibition and probably utilizes well-known monoamine uptake inhibition." 

and

"Weak CLO effect on NMDARs at therapeutic concentrations of 0.5 - 4 µM [36,37] may provide a clue why CLO is rarely utilized for treatment of neuropathic pain in contrast to ATL and DES."

Minor points

Page 3, line 111. I think  “DES” is a mistake,  ATL should be the correct one.  -- Corrected

Page 5, line 166. “Fig 2D” is wrong and should be corrected into “Fig 2E”. -- Corrected

Typos: page 5, line 172 : data “were” compared….. -- Corrected

 page 6, line 202  : “7 mm [Li+]”  instead of “72 mM”. -- Corrected

Reviewer 2 Report

In this manuscript, using patch clamping, Boikov et al. showed that Li+ could increase the IC50 of amitriptyline (ATL), and desipramine (DES) in a calcium-dependent manner. Furthermore, with varied Li+ concentrations, they showed that this effect could happen in the therapeutic range of Li+ concentrations (0.5 - 1 mM). Based on this, the authors proposed that Li+ could weaken the therapeutic effects of ATL and DES during combined therapy. The findings are interesting and the experiments are well conducted.

However, my only concern is that the authors proposed a therapeutic suggestion on the combined therapy of Li+ with Tricyclic antidepressants (TCAs), amitriptyline (ATL), and desipramine (DES). However, there is a lack of clinical data support or correlation with these findings. Besides, further classical depression behavioral studies, such as forced swim test or sucrose preference test, on rats coinjected with Li+ and ATL/DES versus rats just got ATL/DES should be done. With knowledge of these results would make the finding of this paper more convincing and the results more solid.

Author Response

We thank the reviewer for valuable comments. According to the recommendations we have made some changes to the manuscript.

In this study we focused on molecular mechanisms of NMDARs inhibition by tricyclic antidepressants. Based on observed Ca2+-dependent effects of ATL and DES on NMDARs occurring at therapeutic concentrations we have made an assumption on the role of these mechanisms in ATL and DES induced analgesic effects in vivo. Our data do not contradict current clinical data on ATL, DES and CLO usage against the neuropathic pain. We added references 36, 37 to illustrate this point.

We agree that our observation on drug-drug interaction in vitro requires validation by in vivo experiments, which are planned for the next study. In discussion we also emphasized that that observed effects on NMDARs concern only analgesic properties, while antidepressant action is completely different mechanism (page 8 second and last paragraphs).